# SR-TGAN: Smoke Removal with Temporal Generative Adversarial Models in Robot-assisted Surgery

Mengya Xu*
*Department of Electronic Engineering*
*The Chinese University of Hong Kong*
Hong Kong SAR, China
mengyaxu@cuhk.edu.hk

Omer Raza*
*Department of Computer Science*
*Purdue University*
West Lafayette, Indiana, USA
raza12@purdue.edu

An Wang
*Department of Electronic Engineering*
*The Chinese University of Hong Kong*
Hong Kong SAR, China
wa09@link.cuhk.edu.hk

Hongliang Ren[†]
*Department of Electronic Engineering*
*The Chinese University of Hong Kong*
Hong Kong SAR, China
hren@cuhk.edu.hk

*Abstract*—The occurrence of smoke during endoscopic surgery hampers the ease of navigation and obstructs clear visibility, thereby presenting challenges and amplifying risks within surgical procedures. Current image processing-based smoke removal methods predominantly utilize machine learning approaches, specifically Generative Adversarial Networks (GAN). However, they encounter challenges in effectively preserving fine details and generating realistic images, a critical reason being the i.i.d assumption of the inputs. To tackle these issues, we present SR-TGAN: an innovative approach for *Smoke Removal with the Temporal Generative Adversarial Network*. Our model leverages the temporal dynamics inherent in surgical videos to significantly enhance the reconstructed images' quality. Specifically, SR-TGAN integrates sequential contextual information from closely preceding frames to effectively eliminate smoke, especially in regions where inferring the background is challenging such as in a highly occluded region. By comparing our SR-TGAN with the state-of-the-art DeSmoke-LAP, our method exhibits enhanced effectiveness in eliminating smoke from a dataset of 500 test images. Both visual inspection and quantitative metrics support this conclusion. In particular, the JNBM metric exhibits improvement from 1.37 (input images) to 1.49 (DeSmoke-LAP generated images) to 1.51 (SR-TGAN generated images), while FADE decreases from 0.737 to 0.360 to 0.346 for the corresponding image sets. The implications of this study are significant as they have the potential to reduce surgical risks, alleviate surgeons' workload by reducing the need to remove smoke physically, and enhance the precision of other computer vision algorithms utilized in live endoscopic surgeries. The code is available at https://github.com/XuMengyaAmy/SR-TGAN.

*Index Terms*—Smoke Removal, Generative Adversarial Network, Temporal Consistency, Robot-assisted Surgery

This work was supported in part by the Hong Kong Research Grants Council (RGC) Collaborative Research Fund under Grant CRF C4026-21GF; in part by the General Research Fund under Grant GRF 14211420 and GRF 14216022; in part by the STIC Shenzhen-Hong Kong-Macau Technology Research Programme (Type C) under Grant 202108233000303; and in part by the Key Project 2021B1515120035 (B.02.21.00101) of the Regional Joint Fund Project of the Basic and Applied Research Fund of Guangdong Province.
* Co-first authors.
[†] Corresponding Author.

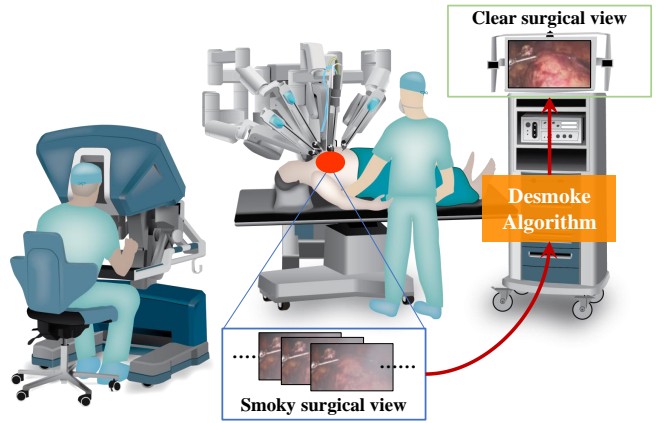

Fig. 1. The presence of smoke during surgical procedures poses a challenge for surgeons, hindering their ability to operate effectively. Desmoking algorithms have been developed to assist in removing smoke, ultimately improving the visibility of surgeons.

## I. INTRODUCTION

Robot-assisted surgery is an advancing field that allows for minimally invasive procedures with improved ergonomics and vision manipulation. However, one issue hampering the widespread adoption of robotic surgery is the presence of surgical smoke produced during common steps such as cautery. A considerable amount of smoke is generated in routine surgical procedures such as laser ablation and electrocautery. Surgical smoke hinders the surgeon's view of the surgical site and makes it difficult for them to navigate the endoscope tubes. Reduced visibility increases the risk of surgical errors and inadvertent injuries. It also poses potential health hazards to

operating room staff through inhalation of smoke particulate matter over prolonged exposure [1]. While smoke evacuation systems have helped, residual smoke still affects the quality of video feeds used by the surgeon for navigation. In such surgical procedures, eliminating or minimizing smoke occlusion from the endoscopy camera feed, where live feedback and quick response is important, requires specialized tools and manual effort and consumes time. A precise, automated and less obstructive smoke removal approach can, therefore, be a valuable asset, allowing for time efficiency and potentially providing a clearer surgical field. Thus, developing effective computational methods for real-time smoke removal from endoscopic video streams can help address this challenge and further improve the capabilities of robotic surgical systems, as illustrated in Fig. 1.

With the availability of high computation power, memory, and parallelization of work via GPUs and cloud resources in the past decade and a half, deep learning approaches to standard computer vision tasks have been increasingly on the rise. Consequently, there have been numerous medical applications that benefit from the use of deep learning models such as image super resolution [2], augmentation & synthesis [3], image-to-image translation tasks [4] and noise removal [5].

The removal of smoke from endoscopic surgical videos has also garnered significant attention, mainly because of its prevalence and significance in medical surgeries. Additionally, the presence of atmospheric modeling and computer vision research has further intensified this focus. To accomplish this goal, multiple deep learning methods have been utilized, such as augmenting datasets and refining model losses and architectures which have achieved varying levels of success. Currently, GAN [6] models have emerged as one of the most prominent networks for achieving smoke removal, often serving as the benchmark. Specifically, CycleGANs [7] have gained popularity in smoke removal due to the impracticality of obtaining a corresponding set of smoky and non-smoky images for training purposes. Despite this distinction, these methods struggle to preserve fine details, generate realistic images and achieve complete smoke removal in highly occluded regions, which limits their practical usability.

An important factor underlying this limitation is that all current deep learning-based methods treat each image input in isolation, the independent and identically distributed (i.i.d) images assumption. This approach overlooks the fact that surgical video frames form a sequence and share a consistent background or have the correlation among the successive frames. As a result, valuable contextual information is lost when analyzing individual images independently. It hence becomes important to incorporate information from previous frames while leveraging existing i.i.d techniques and models to enhance the effectiveness of smoke removal.

Therefore, in this study, we address this issue of the lack of short-term past context by providing the following contributions:

- We propose a novel model, namely SR-TGAN, specifically designed to remove smoke in surgical videos.

In contrast to DeSmoke-LAP, a state-of-the-art smoke removal model [8], our proposed SR-TGAN method incorporates temporal information from successive video frames using the designed attention-based temporal relation network. This enhancement improves smoke removal performance while maintaining computational and storage efficiency.

- We compare our attention-based Temporal Neural Network (TRN), integrated into the generator, with two other commonly used models for utilizing temporal information, namely, 3D Convolution Neural Network (3D CNN) and Long Short-Term Memory (LSTM). Our attention-based TRN achieves a better balance between computational complexity and smoke removal performance. By performing the ablation experiments, and demonstrating flexibility w.r.t where the attention-based TRN is inserted, we show that such attention-based TRN is model-agnostic and can potentially utilized in other architectures.
- We conduct extensive validations on the public benchmark surgical datasets. Our method demonstrates significantly improved performance, outperforming the current leading method substantially.
- We explore our approach's robustness by introducing varying corruption levels to images. The outcomes indicate stable performance as the severity of corruption intensifies.

## II. RELATED WORK

### A. Haze and Smoke Removal

The task of removing smoke in robotic surgery has similarities with the process of dehazing images in computer vision. Existing methods for image dehazing mostly rely on the principles of image-dehazing theory, utilizing atmospheric models, priors, and parameter estimation. Navalel et al. [9] propose an approach that focuses on estimating an atmospheric model using an equation based on the assumption that the density of haze can indicate depth. Subsequently, a simple regressor is trained to determine the parameters in the equation. In contrast, Chunming et al. [10] employ more advanced modeling equations and heuristics, including the use of dark channel (DC) loss and inter-channel (IC) loss. However, relying solely on human heuristics may not adequately address the intricate distribution and edge cases present in smoky images.

Recently alongside, deep learning methodologies have been employed for smoke removal tasks [11]. Several studies have focused on improving the smoke removal dataset and integrating various smoke removal losses into standard model losses. Additionally, researchers have employed convolutional neural networks (CNNs) for the task of transmission map prediction [12]. Fully end-to-end deep learning approaches, such as CycleDehaze [13], have also been explored. CycleDehaze [13] introduces an unsupervised approach that integrates a perceptual loss, similar to a CycleGAN. Another improvement to this work involves using an attention-based mechanism to achieve smooth transitions in non-smoke-heavy regions [14].

The benchmark DeSmoke-LAP [8] model is based on the CycleGAN [7] and enhances the effectiveness of smoke removal by incorporating inter-channel discrepancies and leveraging a dark channel prior loss function. This approach not only aids in eliminating smoke but also prioritizes the preservation of the original semantic information and lighting characteristics of the scene.

However, all current implementations are based on the assumption that the modeling function receives an i.i.d image as input. In the case of real-time surgical image frames, it is important to consider the actual distribution of these frames. By analyzing the surrounding frames with the current keyframe, it may be possible to enhance smoke removal, which is the primary objective of our research. This approach can be particularly effective in image regions where the background cannot be accurately determined due to a significant amount of smoke, necessitating contextual information from preceding video frames.

### B. Unpaired Image-to-Image Translation

There are a lot of real-life scenarios where paired images may not be available, in part due to the impracticality of replicating the exact process flow such as having paired smoky and clear images from live surgical feeds. Additionally, it is not the prime objective of the surgeon to obtain such pairs. In scenarios where paired inputs and outputs, which relate two distinct data domains, are not available, several methods have been proposed. Rosales et al. [15] present an approach based on Bayesian principles that incorporates a prior derived from a patch-based Markov random field of an original image and a probability component obtained from various images with different styles. Instead of directly employing separate models for each domain, recent approaches like Coupled Generative Adversarial Networks [16] and cross-modal scene networks [17] utilize a weight-sharing method. This technique aims to develop a common representation that can be applied across different domains. Liu et al. [18] expand this framework by merging Variational Autoencoders (VAEs) [19] with Generative Adversarial Networks (GANs). Concurrent research [20], [21] promotes a connection between the input and output by encouraging them to have some common "content" characteristics, despite differences in "style". These techniques incorporate adversarial networks and introduce additional components to guarantee that the output closely resembles the input within a specified measurement framework. Examples of such frameworks, as highlighted in various studies ( [20], [21]), include class label spaces, image pixel spaces, and image feature spaces. In our case, we utilize a commonly employed cycle consistency loss, a key component of the CycleGAN architecture, to cater to the unpaired nature of the data.

### C. Robustness

A significant portion of research in computer vision robustness has focused on the substantial challenges posed by adversarial examples designed to mislead models [22], [23]. To evaluate classifier robustness, benchmark datasets have been created for two additional robustness aspects: corruption and perturbation [24]. These datasets facilitate the assessment and validation of robustness improvements across a varied test set, comprising both corrupted and perturbed images [25], [26]. In our research, we generate a new test dataset through the application of corruption and perturbation techniques, which allows us to evaluate the robustness of our method.

## III. METHODOLOGY

### A. Preliminaries

The Cycle Generative Adversarial Networks (Cycle-GAN) [7] model, widely utilized in various evaluated benchmarks on datasets and its aptness for unpaired data training, serves as the backbone of our study.

*1) GAN:* Before discussing CycleGAN, we briefly discuss the regular Generative Adversarial Network (GAN) [6] which is composed of a generator and discriminator network. The training process involves training a generator to generate false data, and simultaneously training a discriminator network to differentiate between the generator's artificially created data and genuine examples. If the discriminator quickly detects the fabricated data generated by the generator, the generator faces a penalty. Through the continuous feedback loop between these adversarial networks, the generator gradually improves its output, generating higher-quality and more convincing results, while the discriminator becomes more adept at identifying artificially constructed data. Ultimately, the GAN model can produce highly realistic images.

*2) CycleGAN:* Similar to GAN [6], CycleGAN [7] consists of generator and discriminator networks. However, unlike GAN, CycleGAN consists of two generators and two discriminators which transform images between the two domains, smoky and smoke-free images in our case. They also employ adversarial and cycle-consistency losses to achieve unpaired image-to-image translation between the source $X$ and target $Y$ domains.

*3) DeSmoke LAP:* We build our SR-TGAN model based on the DeSmoke LAP [8] and incorporate the temporal relation network into the generator while maintaining identical configurations to ensure fairness in our comparison.

DeSmoke-LAP [8] is based on the CycleGAN and employs a blend of heuristics, perceptual, and adversarial losses to eliminate smoke effectively. The loss equation is as follows:

$$\mathcal{L}_{loss} = \mathcal{L}_{cyc} + \mathcal{L}_{adv} + \lambda * \mathcal{L}_{idt} + \mathcal{L}_{DC} + \mathcal{L}_{IC} \quad (1)$$

where $L_{adv}$, $L_{idt}$, $L_{cyc}$, $L_{DC}$, $L_{IC}$ are the adversarial, identity, cyclic consistency, dark channel prior and inter-channel discrepancies loss functions, respectively [8]; $\lambda$ is a hyperparameter. The purpose of these losses is to eliminate smoke from the surgical scenes while preserving the scene's inherent structure, including the background, and lighting.

### B. SR-TGAN

Our SR-TGAN consists of the unpaired image-to-image cycle-consistent generative adversarial network (Cycle-GAN) [7] and the attention-based temporal relation network

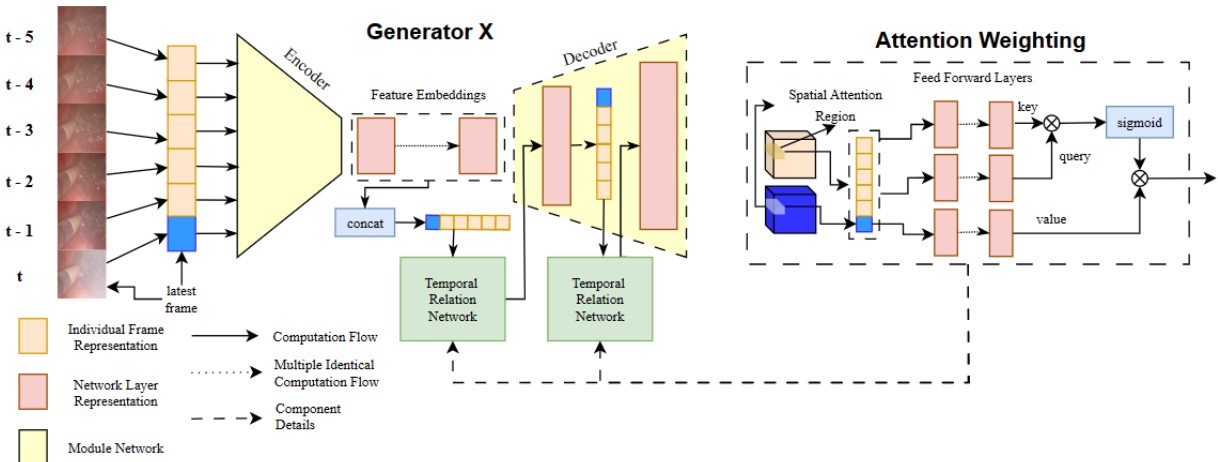

Fig. 2. Architecture of the SR-TGAN. SR-TGAN is composed of 2 generators and 2 discriminators, which is similar to CycleGAN [7]. However, the generator $G_x$ of our SR-TGAN integrates the attention-based temporal relation network (TRN) at both the encoder and decoder layers. The attention-based TRN captures temporal information from the preceding image sequences by utilizing the self-attention mechanism over a small spatial region.

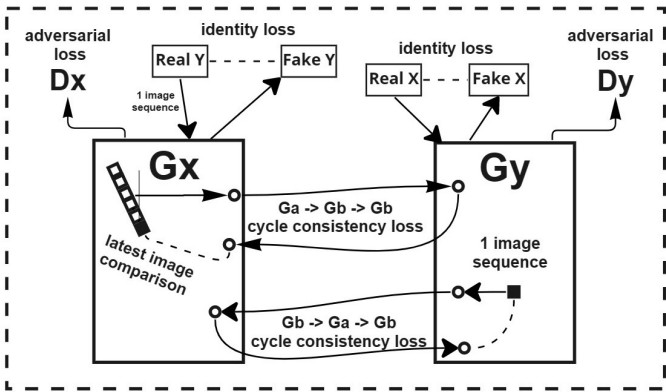

Fig. 3. CycleGAN losses in our SR-TGAN involve explicit sequence indexing juxtaposed with generator and discriminator modules.

(TRN) by incorporating sequential context information from adjacent frames, as shown in Fig. 2. Our attention-based TRN can be integrated into the generator at two stages: after the encoding process (SR-TGAN-EN) and during the intermediate decoding phases (SR-TGAN-DE).

The goal of our SR-TGAN is to create a mapping between unpaired smoky images ($X$) and clean images ($Y$). To achieve this, the network incorporates two discriminators. The first discriminator $D_X$ is employed to differentiate between clean real images $X$ and the generated clean images $G_X(X)$. The second discriminator $D_Y$, is tasked with distinguishing between smoky images and the generated smoky images $G_Y(X)$. These two pairs of generators and discriminators work in tandem in a zero-sum game to iteratively improve generated image quality

Besides this conventional configuration, we can capitalize on the temporal dynamics in the surgical videos to improve image quality.Therefore, In our attention-based TRN, we perform an attention-based weighting channel-wise across a small

spatial region over the short sequence of images. The locality of the spatial region enables us to share weights, reduce model parameter size, be image size agnostic, and facilitate reliable convergence. The process of attention-based TRN involves working with three groups of vectors: queries (Q), keys (K), and values (V). This involves computing a weighted sum of the value vectors, determined by the similarity between the query and key vectors. The scaled dot-product attention can be formally expressed as follows:

$$Attention(Q, K, V) = softmax(\frac{QK^T}{\sqrt{d}})V \qquad (2)$$

Given a series of consecutive images $X = x_{t-l}, ..., x_{t-1}, x_t$ that contains the current frame $x_t$ and a set of its previous frames, the generator network $G_x$ encodes these images into $l$ encodings $F = f_{t-l}, ..., f_{t-1}, f_t$. After the encoder, our designed attention-based temporal relation network (TRN) can be used to obtain the temporal information from a set of encodings $F = f_{t-l}, ..., f_{t-1}, f_t$ through the self-attention mechanism employed in the Transformer [27]. In this scenario, the queries, keys, and values are acquired through linear projection of these encodings. Consequently, the attention operator in our TRN can be defined as

$$TRN(F) = Attention(W_q F, W_k F, W_v F) \qquad (3)$$

where $W_q, W_k, W_v$ are linear projection transforms of the encodings.

The feature representation, corresponding to the latest frame $f_t$, is treated as the Query (Q). The feature encodings $F = f_{t-l}, ..., f_{t-1}, f_t$ are considered as keys (K). We end up with a weighted value encoding for the color channels over the corresponding local spatial region. This makes sense since there is not much movement between successive frames. However, this formulation does imply a lack of global spatial context. On the other hand, this also makes the network very

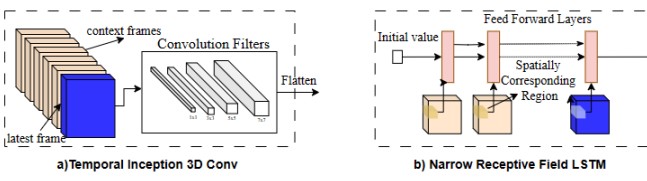

Fig. 4. Two additional implementations of the temporal relation network(TRN): (a) 3D Convolution Neural Network and (b) narrow receptive field LSTM.

lightweight, resulting in just a 3% increase in model size. The same attention process occurs at the decoding layer, with the key frame being the weighted attention-based vector, conditioned on the latest frame from the encoding stage.

### C. Objective Function

The standard losses and customized losses, such as $L_{DC}$ and $L_{IC}$, have been slightly modified to account for the losses incurred on the latest image frame, $x_t$ only. Specifically, the cycle consistency loss $L_{cyc}$ for the generator now includes a perceptual comparison with the latest frame, index $t$. This modification encourages the generator to exclusively focus on the newest frame as the primary frame for generative comparison. A similar adjustment has been made for the identity loss $L_{idt}$. Fig. 3 shows that the final objective functions are similar to (1) except for the fact that frame $x_t$ is first explicitly extracted and then used for loss computation. For instance, the Cyclic loss for forward transformation will be as follows:

$$L_{cyc} = |G_B(G_A(x_{t...t-l})_t) - x_t| \tag{4}$$

A similar formulation holds for the Identity loss, $L_{idt}$.

## IV. EXPERIMENTS

### A. Dataset

**Laparoscopic Hysterectomy Dataset** includes a total of 10 videos for training and cross-validation purposes. Additionally, there are 500 sequential images from 10 separate videos which are used exclusively for testing purposes [8].

These videos are divided into frames at a rate of 1 frame per second (fps). Each video yielded 300 clear images and 300 hazy images, resulting in a dataset consisting of 3000 clean images and 3000 images with smoke. All images are resized into $720 \times 540$.

**Transoral robotic surgery (TORS) Dataset** contains 10 videos. Each video contains approximately 150 clear images and 40 smoke images. The entire data set contains 1697 clear images and 370 images with smoke events. We use this dataset for training and cross-validation purposes.

### B. Implementation Details

*1) Network Architecture:* To implement the SR-TGAN, we utilize standard AlexNet-style CNNs as the discriminators, each initialized with 64 filters. We utilize a Unet-256 model for the generator with 48 initial filters. LSGAN [28] loss is utilized to reduce the saturation effect instead of the vanilla loss [6]. Additionally, the DC loss and IC loss are incorporated [8].

*2) Training Details:* A learning rate of 0.0002 is used while a batch size of 16 for the CycleGAN and 2 for SR-TGAN is employed, as the SR-TGAN receives a sequence. Finally, as practical measures to reduce the probability of model collapse and stabilize training, noise is incorporated into both the input images and the intermediate images produced during the generation process, alongside incorporating random cropping and linear learning rate decay. To better accommodate varying numbers of consecutive images during the deployment phase, we opt for the random sequence length strategy during training, ranging from 1 to 6 frames, instead of using the fixed sequence length strategy. This approach allows for more flexibility and less computational cost. In the extreme case where the sequence length during the deployment phase is set to 1, our SR-TGAN can also achieve the task without relying on temporal information.

*3) Data Assistance:* During the training process, we mix the Laparoscopic Hysterectomy Dataset and the in-house TORS Dataset because the in-house TORS Dataset can provide richer temporal information, such as longer image sequences and more frequent smoke events, for model learning.

### C. Evaluation Metrics

As our data is unpaired and obtained from real robotic surgeries, we cannot attain paired ground-truth images of clear and hazy versions. To assess the quality of the produced images, we depend on various image quality metrics that do not require a reference image, namely the just noticeable blur metric (JNBM) [29] and fog aware density evaluator (FADE) [30]. This is consistent with the metrics outlined in DeSmoke-LAP [8]. In general, improving image quality, particularly regarding smoke removal, should lead to an increase in JNBM and a decrease in FADE.

*1) Just noticeable blur metric (JNBM):* evaluates the image's perceived sharpness, with a higher number indicating greater sharpness. It specifically examines how the human eye and visual processing system respond to sharpness at varying levels of contrast and captures the level of blurriness present in the image's edges.

*2) Fog aware density evaluator (FADE):* is employed to assess the density of fog within the image. A higher number indicates a greater amount of fog covering the image. This algorithm is designed using the principles of natural scene statistics (NSS) and includes features derived from statistical analysis that are sensitive to the presence of fog.

*3) Floating point operations (FLOPs) and Parameters:* are used to evaluate the speed and size of neural networks in inference operations.

## V. RESULTS

A total of 500 test images from the Laparoscopic Hysterectomy Dataset are used to evaluate the performance of the approaches. They are fed to the trained DeSmoke-LAP model [8] and our trained SR-TGAN model, resulting in 500 corresponding output images for each model. We provide test

TABLE I

DESMOKING RESULTS AND INFERENCE SPEED USING DIFFERENT APPROACHES. OUR SR-TGAN ACHIEVES A BETTER BALANCE BETWEEN SMOKE REMOVAL PERFORMANCE AND INFERENCE SPEED.

| Approach | | | Desmokeing performance | | Inference speed | |
|---|---|---|---|---|---|---|
| | | | JNBM ↑ | FADE ↓ | FLOPs (G) ↓ | Parameters (M) ↓ |
| Original Input | | | 1.370 | 0.737 | N/A | N/A |
| Non-Temporal | | DeSmoke-LAP | 1.490 | 0.360 | 100.7 | 28.3 |
| Temporal | Heavyweight | CycleGAN+3D CNN | 1.530 | 0.342 | 1532.0 | 1045.1 |
| | Lightweight | CycleGAN+NRF LSTM | 1.480 | 0.383 | 243.2 | 28.4 |
| | | SR-TGAN-EN | 1.500 | 0.355 | 197.4 | 28.4 |
| | | SR-TGAN-DE | 1.510 | 0.347 | 224.9 | 28.5 |
| | | SR-TGAN | **1.510** | **0.346** | 333.7 | 28.7 |

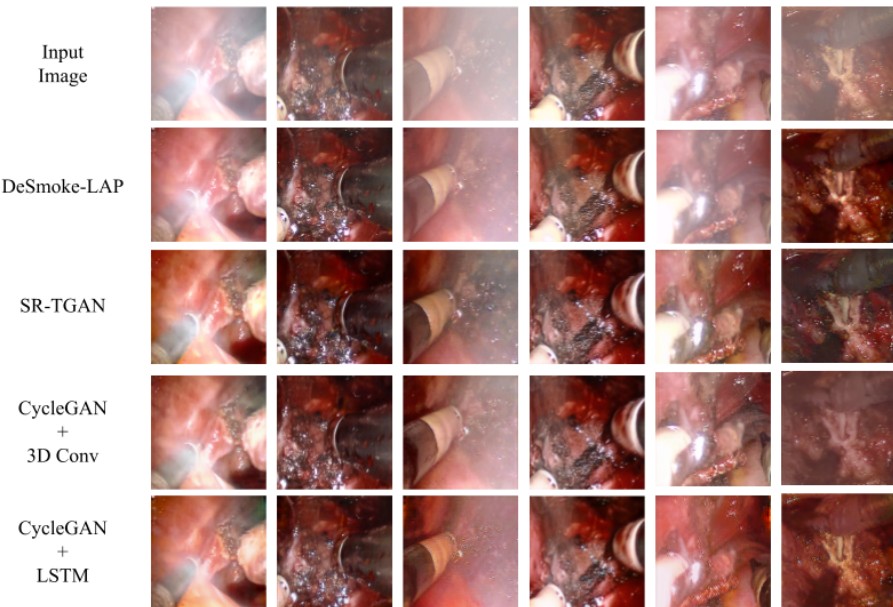

Fig. 5. SR-TGAN harnesses the power of past image sequences to achieve a remarkable enhancement in generated images compared to DeSmoke-LAP. As a result, the generated images not only excel in smoke removal but also stand out in terms of color preservation and maintaining structural integrity.

TABLE II

THE CROSS-VALIDATION RESULTS ARE OBTAINED FROM BOTH THE STATE-OF-THE-ART DESMOKE-LAP AND OUR PROPOSED SR-TGAN.

| | fold 1 | | fold 2 | | fold 3 | | fold 4 | | fold 5 | | Mean | |
|---|---|---|---|---|---|---|---|---|---|---|---|---|
| | JNBM ↑ | FADE ↓ | JNBM ↑ | FADE ↓ | JNBM ↑ | FADE ↓ | JNBM ↑ | FADE ↓ | JNBM ↑ | FADE ↓ | JNBM ↑ | FADE ↓ |
| DeSmoke-LAP | 1.38 ± 0.14 | 0.38 ± 0.14 | 1.37 ± 0.14 | 0.41 ± 0.13 | 1.38 ± 0.16 | 0.42 ± 0.13 | 1.32 ± 0.13 | 0.45 ± 0.13 | 1.42 ± 0.12 | 0.37 ± 0.09 | 1.37 ± 0.14 | 0.41 ± 0.12 |
| SR-TGAN | 1.31 ± 0.12 | 0.34 ± 0.07 | 1.43 ± 0.13 | 0.41 ± 0.14 | 1.44 ± 0.15 | 0.42 ± 0.13 | 1.43 ± 0.15 | 0.42 ± 0.13 | 1.32 ± 0.11 | 0.34 ± 0.07 | **1.39 ±0.13** | **0.39 ± 0.11** |

image sequences with a uniform length of 6 to our SR-TGAN model.

As shown in Fig. 4, we also combine two other common strategies for handling sequence images with generators to incorporate temporal information for smoke removal: (1) CycleGAN+3D CNN: Inception network-based 3D Convolution Neural Network [31] is applied to cover a range of receptive fields across the different frames. However, this significantly increases the size of the generator by 40 times compared to the original and leads to slower real-time inference. (2) CycleGAN+LSTM: a narrow receptive field LSTM mechanism is employed to make the model more lightweight. By analyzing Table I, which includes the metric from input images,

DeSmoke-LAP [8], CycleGAN+3D CNN, CycleGAN+LSTM, SR-TGAN-EN, SR-TGAN-DE, and SR-TGAN, the following findings are observed: (1) All methods achieve higher JNBM and lower FADE than the input smoke image with JNBM and FADE of 1.370 and 0.737, respectively. This shows that these methods have the effect of removing smoke. (2) The JNBM of SR-TGAN achieves a 0.02 increase compared to the JNBM of DeSmoke-LAP, and the FADE of SR-TGAN is 0.3 less than that of DeSmoke-LAP. This demonstrates that our SR-TGAN model effectively enhances the smoke removal effect by utilizing temporal information. (3) Compared with SR-TGAN-EN and SR-TGAN-DE, which integrates the attention-based TRN only after the encoder or only in the intermediate

TABLE III
JNBM GAIN WITH INCREASING INPUT SEQUENCE LENGTH FOR SR-TGAN ON THE TEST SET.

| SR-TGAN | Seq length 1 | Seq length 2 | Seq length 3 | Seq length 4 | Seq length 5 | Seq length 6 |
|---|---|---|---|---|---|---|
| JNBM ↑ | 1.40 ± 0.12 | 1.45 ± 0.13 | 1.48 ± 0.13 | 1.48 ± 0.13 | 1.49 ± 0.13 | 1.51 ± 0.13 |

TABLE IV
THE JNBM METRIC VALUES FROM DESMOKE-LAP AND OUR SR-TGAN
WERE EVALUATED ON THE CORRUPTED TEST SET FOR ROBUSTNESS
EVALUATION, WHICH CONSISTED OF 5 DIFFERENT CORRUPTION TYPES
AND 3 SEVERITY LEVELS.

| Model | Defocus | Gauss | Zoom | Elastic | Brightness | Mean |
|---|---|---|---|---|---|---|
| DeSmoke-LAP | 1.40 | 1.39 | 1.30 | 1.22 | 1.45 | 1.35 |
| CycleGAN+3D CNN | 1.47 | 1.43 | 1.35 | 1.29 | 1.62 | 1.43 |
| CycleGAN+LSTM | 1.36 | 1.40 | 1.32 | 1.18 | 1.49 | 1.35 |
| SR-TGAN | 1.42 | 1.4 | 1.33 | 1.24 | 1.49 | 1.38 |

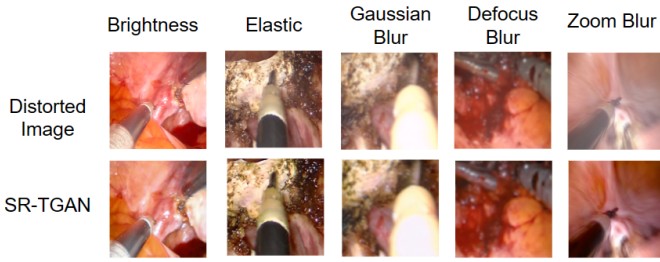

Fig. 6. Corrupted images, alongside their corresponding model outputs, were acquired to highlight the robustness of the system. Each distortion type has the designated severity level of 3, medium degree.

decoding stage separately, our SR-TGAN, which inserts the attention-based TRN in two locations, obtains the best results. Results suggest that inserting the attention-based TRN after the encoder only provides little help. However, inserting the attention-based TRN at the decoding layer provides a significant difference, with JNBM being 1.510 and FADE value being 0.347. This makes sense as the decoding layer is more related to perceptual information, while the encoding layer is more concerned with semantic information. We infer that further modifications at the decoding layer can be potentially helpful, while encoder layer modifications may not cause a difference or diminishing gains. (4) Compared to our SR-TGAN, CycleGAN+3D CNN model yields a slightly better JNBM value of 1.530 and FADE value of 0.342. CycleGAN+LSTM model yields a worse JNBM of 1.48 and FADE of 0.383. Our attention-based TRN can provide better gains than the GAN+3D CNN model regarding FLOPs and Parameters while potentially scaling better to wider receptive fields.

In addition to providing quantitative evidence of improved smoke removal through the use of JNBM and FADE, we also present the generated image from the approaches from a visual standpoint, as shown in Fig. 5. These images exhibit enhanced smoke removal and improved structural integrity in heavily occluded areas. To avoid the limitations and particularities of fixed partitioning of data sets and eliminate the adverse effects caused by unbalanced data partitioning in a single partition, our proposed SR-TGAN model is also compared with the cur-

rent state-of-the-art method, DeSmoke-LAP [8], by utilizing a 5-fold cross-validation procedure, as demonstrated in Table II. Analyzing the mean value, it becomes evident that our method exhibits superior performance compared to DeSmoke-LAP, highlighting its effectiveness and superiority.

### A. Robustness

To assess the robustness of a model quantitatively, we can examine its ability to withstand intentional corruption and perturbation of images [26]. This study introduced 5 types of corruption: Defocus Blur, Gaussian Blur, Zoom Blur, Elastic, and Brightness. These corruptions correspond to various types of distortions caused by smoke, such as changes in brightness or movement, namely zoom blur and elastic distortions. For each corruption type, we generate images from 3 different severity levels. These corrupted images are utilized to evaluate the robustness of different models. A model can be considered more robust if it maintains its performance as corruption severity increases. The resulting evaluation is depicted in Table IV. Observing the mean value of 5 different corruption types with 3 severity levels, our SR-TGAN demonstrates greater robustness compared to DeSmoke-LAP and Cycle-GAN+LSTM. While it is not as robust as CycleGAN+3D CNN, it significantly outperforms CycleGAN+3D CNN in terms of inference speed, achieving a favorable balance between task performance and speed. Fig. 6 presents the model output on corrupted images.

### B. Ablation Study

Our SR-TGAN model architecture has the flexibility to accommodate sequences of input images of any size. In the Table III, the JNBM [29] and FADE [30] metrics are computed on the same global test as used for Table I. However, the inputs' lengths vary from 1, essentially a normal i.i.d Cycle-GAN at this point, to 6 for each run of the test dataset. The performance of SR-TGAN on the test set shows a significant rise in the JNBM score as the input sequence length increases. This indicates that longer input sequences positively contribute to the JNBM achieved by SR-TGAN. It can also be observed there are indeed diminishing gains w.r.t increasing sequence length. Thus, there can be a reasonable compromise on the choice of sequence length to be used in real-time inference.

Investigating whether there are any noticeable benefits in increasing the sequence length is important. This analysis is heavily influenced by the nature of the dataset and the specific objective being addressed. In the case of surgical videos at a given frame rate, this examination can aid in identifying an optimal balance between smoke removal performance and computational resources expended.

## VI. CONCLUSION

We propose SR-TGAN, a novel method for smoke elimination in surgical videos captured during robot-assisted procedures by utilizing temporal information from consecutive video frames. Our SR-TGAN is based on a cycle-consistent generative adversarial network (CycleGAN) and integrates the attention-based temporal relation network. The model effectively eradicates smoke by assimilating contextual information from the preceding frames, especially in areas where the image background is arduous to infer. Quantitative, such as JNBM and FADE, and qualitative, particularly visual, analyses were employed to compare our SR-TGAN model with the state-of-the-art DeSmoke-LAP method [8], the 3D CNN-based CycleGAN and LSTM-based CycleGAN models. Our SR-TGAN model achieves an optimal balance between smoke removal performance, by outperforming all but the heavy 3D CNN model, and inference speed, by being faster than the 3D CNN model and similar in speed to the rest.

Future research could incorporate complex attention modules and geometrically aligning constraints, such as Mutual Information loss. The proposed attention-based temporal relation network, being model agnostic, can be used in different layers with little to no modifications. It can also be replicated in more powerful models like large diffusion models [32] to achieve superior smoke removal performance. Another line of work could involve investigating a balance between model size alongside input sequence length and inference speed during real-time surgical video live feed to obtain the most pragmatic configuration. This approach would uphold structural integrity and maintain accurate background information in real-time automated processes.

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
