# OpenReview forum: "SR-TGAN: Smoke Removal with Temporal Generative Adversarial Models in Robot-assisted Surgery"
_IEEE.org/EMBS/BHI/2024/Conference — IEEE BHI'24_

### Official Review · Reviewer_RJsu · 2024-08-09
**Review 132**

**Overall Rating:** 7
**Confidence:** 3

**Other Quality Metrics:**

(a) Clarity of writing: good
(b) Clinical Significance: good
(c) Methodological Novelty: good
(d) Experiments and Results: good

**Questions For The Authors:**

As for the data, are the two datasets freely available? Moreover, why is it necessary to mix the two datasets as described in the "data assistance" section?

**Strengths:**

Overall, the manuscript is clear, and the methodology is well explained by highlighting first the general meaning of the process (e.g. the general architecture of the GAN) and then going in depth by describing the steps performed and the introduced novelty.

**Summary Of The Paper:**

The work proposed a novel model, namely SR-TGAN for smoke removal in surgical videos. Since the problem of the occurrence of the smoke during endoscopic surgery is a big issue often causing surgical errors, desmoking algorithms and/or deep learning-based methods have been developed in recent years for smoke removal. The proposed SR-TGAN method is designed to incorporate temporal information derived from successive video frames using the designed attention-based temporal relation network rather than analyzing individual images independently (a strategy that is often used but has disadvantages as well as the fact that video frames form a sequence and share a consistent background rather than the single image taken independently).

**Weaknesses:**

As for the paper structure, I suggest to provide a separate Results section, as well as a discussion section. Moreover, figures and tables should apper after the relevant text.

---

### Official Review · Reviewer_dpox · 2024-08-10
**review for paper ID 132**

**Overall Rating:** 6
**Confidence:** 3

**Other Quality Metrics:**

(a) Clarity of writing :good
(b) Clinical Significance  : good
(c) Methodological Novelty : good
(d) Experiments and Results : good

**Questions For The Authors:**

Good paper for the conference

**Strengths:**

The paper proposes new ideas for attention-based deep learning approach for smoke elimination for endoscopic imaging and operation systems.
The paper is well-structured, methodologically sound, and effectively communicated.
The paper clearly states its research questions and objectives, providing a solid foundation for the study.
The image quality is highly adequate and readable.

**Summary Of The Paper:**

The authors proposed the Generative Adversarial Network based approach for smoke removal. The suggested Smoke Removal with the Temporalbased Generative Adversarial Network (SR-TGAN), is a novel method for smoke elimination in surgical videos captured during robot-assisted procedure by utilizing temporal information from consecutive video frames. The method was based on a cycle-consistent generative adversarial network (CycleGAN) and integrates the attention-based temporal relation network.

**Weaknesses:**

The manuscript should be revised in writing to make the readers follow the context more easily. The whole text should be revised for grammar, typos, and extra spaces.

---

### Official Review · Reviewer_VE3Z · 2024-08-11
**Smoke removal using Temporal GAN**

**Overall Rating:** 7
**Confidence:** 5

**Other Quality Metrics:**

(a) Good
(b) Good
(c) Good
(d) Good

**Questions For The Authors:**

1. The paper does not deeply explore edge cases where the model might struggle, such as in extremely dense smoke or rapidly changing surgical environments.
2. While future research directions are mentioned, try to be more specific, outlining clear pathways for integrating the model with existing surgical systems or expanding its capabilities beyond smoke removal.

**Strengths:**

1. Innovative Use of Temporal Information

2. Robust Performance: It shows improved performance in smoke removal, both quantitatively and qualitatively compared to state-of-the-art methods.

3. Efficiency: The model maintains a good balance between performance and computational requirements, making it suitable for real-time surgical applications.

4. Comprehensive Evaluation

**Summary Of The Paper:**

This paper presents a novel approach called SR-TGAN (Smoke Removal with Temporal Generative Adversarial Network) to improve visibility in robot-assisted surgical procedures by effectively removing smoke from endoscopic video feeds. SR-TGAN leverages temporal information from successive frames in surgical videos to enhance the quality of reconstructed images. The model integrates an attention-based Temporal Relation Network (TRN) within a CycleGAN framework to improve smoke removal performance while maintaining computational efficiency. The paper compares SR-TGAN with existing methods using metrics like JNBM and FADE to demonstrate better performance in smoke removal and robustness against image corruption.

**Weaknesses:**

1. Great work for the unpaired dataset however testing the model on a paired GT dataset would be more insightful in providing a real-time comparison and reliability.
2. Robustness evaluation (Table IV) should show comparisons among other models mentioned in Table II for a better performance evaluation.
3. If possible additional metrics or qualitative assessments, such as surgeon feedback, could provide a more comprehensive evaluation.

---

### Decision · Program_Chairs · 2024-09-23

Accept